# FRESCO: FEDERATED REINFORCEMENT ENERGY SYSTEM FOR COOPERATIVE OPTIMIZATION

**Nicolas M. Cuadrado, Roberto A. Guillén & Martin Takáč**
Department of Machine Learning
Mohamed Bin Zayed University of Artificial Intelligence
Masdar City, Abu Dhabi, UAE
`{nicolas.avila, roberto.guillen, martin.takac}@mbzuai.ac.ae`

## ABSTRACT

The rise in renewable energy is creating new dynamics in the energy grid that promise to create a cleaner and more participative energy grid, where technology plays a crucial part in creating the required flexibility to achieve the vision of the next-generation grid. This work presents FRESCO, a framework that aims to ease the implementation of energy markets using a hierarchical control architecture of reinforcement learning agents trained using federated learning. The core concept we are proving is that having greedy agents subject to changing conditions from a higher level agent creates a cooperative setup that will allow for fulfilling all the individual objectives. This paper presents a general overview of the framework, the current progress, and some insights we obtained from the recent results.

## 1 INTRODUCTION

Climate change has emerged as one of our planet's most pressing challenges. It is causing various environmental, economic, and social impacts, including rising sea levels, more frequent extreme weather events, and threats to global food security IPCC (2018). Promoting renewable energy sources helps reduce greenhouse gas emissions from the energy sector. It often involves using microgrids and small-scale power systems that can operate independently or in conjunction with the main grid. Hierarchical control systems ensure microgrids' efficient and effective operation by coordinating the various components and managing the energy flow Yang et al. (2020). There are privacy concerns associated with microgrids, such as collecting and using data about energy consumption, and addressing these issues proactively is essential to ensure the benefits while protecting the privacy of individuals and communities Tyav et al. (2022). In this work, we present a hierarchical control framework composed of reinforcement learning (RL) agents, including federated learning (FL), to enable scaling without risking the information from participants. The ultimate goal is to reduce the collective carbon footprint of a group of microgrids. We called this framework FRESCO.

## 2 METHODS

FRESCO comprises three components: 1) An OpenAI Gym environment that represents different microgrid setups, 2) a hierarchical structure of RL agents that control the microgrid at different scales, and 3) a training stage using FL. We modeled this framework to represent the real case scenario in which agents tend to pursue individual objectives, namely, are greedy; however, agents in the same label end up following a common goal dictated by a higher layer agent. Figure 1 presents a general view of our research scenario.

Our environment generates synthetic data for any microgrid scenario from a configuration detailed in the appendix. We searched for the appropriate RL methodology for each layer's objective. Layer 1 agents will seek to reduce their energy bill by managing a storage system at home. In Layer 2, the agents define trading prices in a microgrid, aiming to reduce the carbon footprint by promoting energy exchange within neighboring houses. Finally, layer 3 will encourage energy exchange between microgrids by setting those prices and ensuring that the distribution system's physical constraints are respected. We propose a metric that evaluates two scenarios: one without batteries and

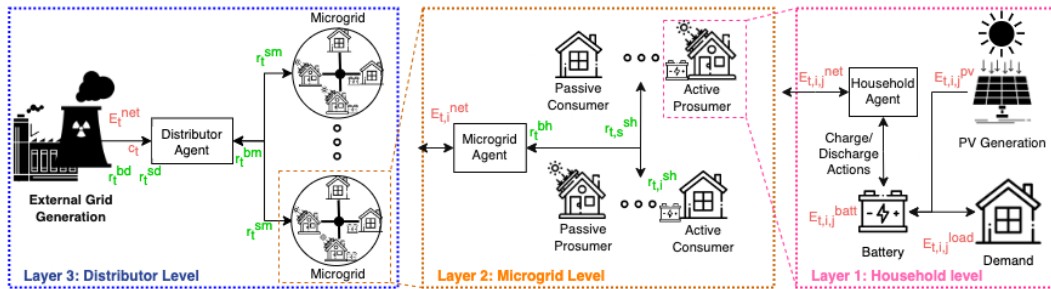

Figure 1: Three-layer architecture of our hierarchical microgrid control

|  | CVXPY | A2C - No FL | FRESCO |
|---|---|---|---|
| Train reward | -0.915 | **-0.735** | -0.81 |
| Train price score | **-0.103** | -0.097 | -0.10 |
| Train emission score | **-0.223** | -0.1522 | -0.18 |
| Test price score | -0.0889 | -0.083 | **-0.099** |
| Test emission score | -0.19 | -0.19 | **-0.28** |

Table 1: Average performance of households. Lower is better for all except reward.

one with FRESCO. It assesses the system's overall performance measuring the contribution of each household individually, and aggregates the results at the microgrid and distributor levels.

$$P_{delta} = P_{base} - P_{FRESCO}, \qquad (1)$$
$$C_{delta} = C_{base} - C_{FRESCO}. \qquad (2)$$

## 3 EXPERIMENTS AND RESULTS

We completed the training for Layer 1 agents using our version of the Advantage Actor-Critic algorithm (A2C), which considers causality and microgrid attributes. These agents' actions are defined in the $[-1, 1]$ range, where $-1$ means fully discharging the battery, and $1$ means fully charging it. We trained the agents to be greedy in their objective of reducing bill price at the same time that they can adapt to external stimuli, like changes in the energy price or local energy availability coming from neighboring houses. The training set consists of 6 houses, the validation set of 6 other houses, and the testing set of 10 new houses. Every 200 episodes, the neural network parameters for both the actors and critics are synchronized using the FedAVG approach introduced by McMahan et al. (2016). The baseline to compare the performance of FRESCO is the values obtained when solving the same case with a linear solver. We utilized CVXPY, a linear solver, to establish the theoretically best possible result for the given scenarios. The results in table 1 demonstrate better results than the standard RL approach. In the appendix, Figures 3 and 2 show more detail about the training process.

## 4 CONCLUSION

FRESCO framework uses federated reinforcement learning (FRL) to promote communication and cooperation among smart grids, which tend to be decentralized and must keep users' personal information private. The framework proposed in this paper considers external grid $CO_2$ impact and prices, temperature, different distributed energy resources, and the diverse attributes of households. The approach enables stakeholders to trade energy optimally and in an automated manner without sharing consumer energy data. Results show that FRL can quickly adapt to grid changes and improve optimal policy convergence with a small hit in general training speed. Our approach applies to multiple interconnected microgrids and benefits all stakeholders by lowering energy bills, reducing electricity transmission rates, decreasing $CO_2$ impact, and increasing participation in the energy market.

URM STATEMENT

The authors acknowledge that at least one critical author of this work meets the URM criteria of the ICLR 2023 Tiny Papers Track.

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

## A  APPENDIX

### COMPARISON RL VS. FL

Figures 2 and 3 show the training and evaluation score function through the epochs. Each color stripe represents a microgrid configuration, six in total. The agents go through them twice to test their adaptation capability.

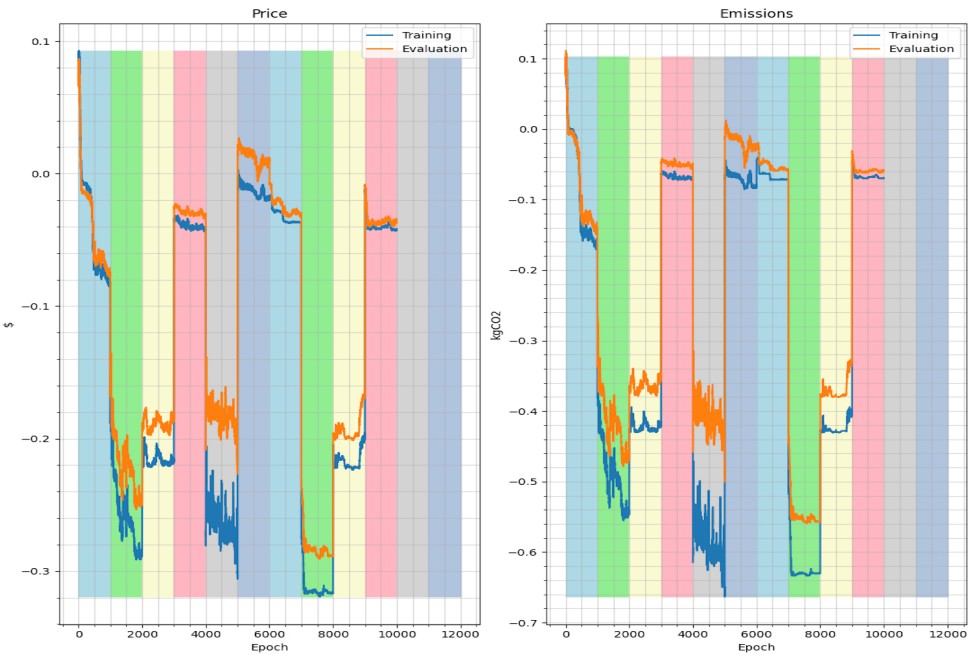

Figure 2: Reinforcement learning A2C run

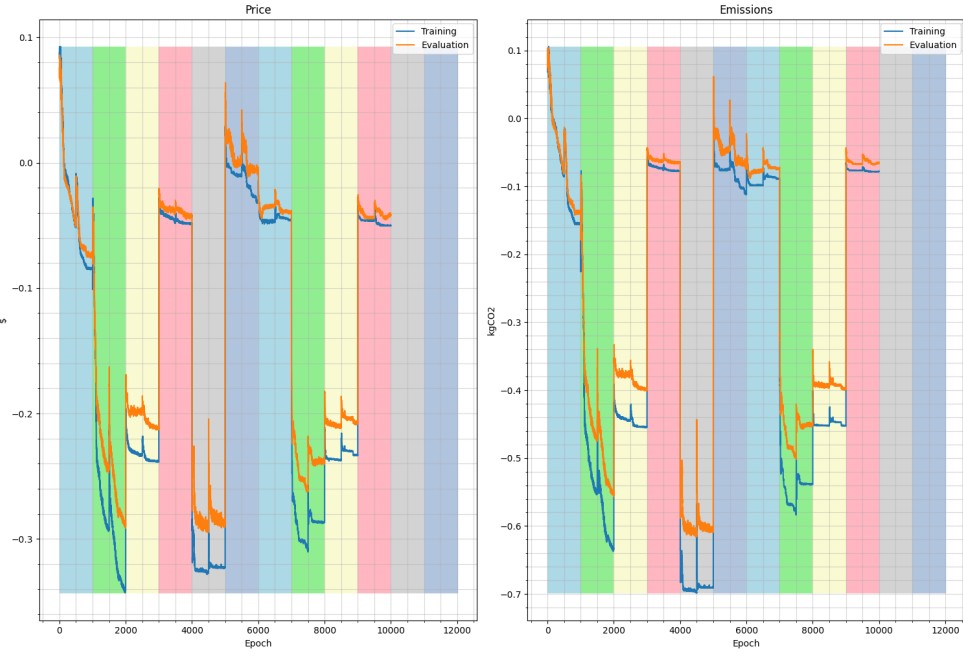

Figure 3: Federated Learning Run

## DATASET GENERATION

We propose to build a data generator that synthetically creates time series data for solar panel generation, residential load, energy price, and energy emissions. Our data generator has several advantages over using real or existing datasets. First, it reduces the dependency on finding or collecting ideal datasets that match our problem setup and objectives. Secondly, we can control the level of stochasticity and variability in the data, which enables us to test different scenarios and assumptions. Finally, it opens the door to using optimizers as the reference point for evaluating the performance of RL algorithms since we can generate data with known optimal solutions. We considered min-max scaling for each type of data we generated by restricting all the variable values to the range $[0, 1]$. For all the cases, the number of generated data points is defined by the maximum number of time steps $T$, defined before. A brief description of the generated is presented next.

### PV GENERATION

The data from solar panel generation belongs to the Layer 1 of our scenario, as it is a DER belonging to the households. It is modeled by creating a base from a rectified $\sin$ function tunned to reach its peak after 12 steps, simulating the peak of solar energy on a clear sky day. It is scaled using the parameter $E_{i,j}^{\text{pv,max}} \in [0, 1]$ and receives perturbation from a noise component modeled as a normal distribution $\mathcal{N}_{\text{pv}}(\mu_{pv}, \sigma_{pv})$, where we defined by default $\mu_{pv} = 0$ and $\sigma_{pv} = 0.1$, but it can be changed as needed.

### RESIDENTIAL LOAD

Also belonging to Layer 1, we defined three types of profiles for this kind of data: family, teenagers, and home business. Depending on the type, we established arbitrary 24-hour base profiles representing each case's expected consumption. The load is parametrized by $E_{i,j}^{\text{load,max}} \in [0, 1]$ that defines the peak consumption. We assume a permanent constant consumption proportional to the peak load denoted as $E_{i,j}^{\text{load,c}}$ with a default value of 0.2 (20% of the peak load is constantly consumed). We add a noise component to the constant consumption, modeled again as a normal distribution $\mathcal{N}_{\text{load}}(\mu_{\text{load}}, \sigma_{\text{load}})$, with $\mu_{\text{load}} = 0$ and $\sigma_{\text{load}} = 0.01$ by default.

### ENERGY PRICE AND EMISSIONS

This kind of data belongs to Layer 3, being external inputs influenced by the availability of generators, market regulations, policy changes, unexpected events, etc. We assume that the grid can always supply any demand, so the relevant aspects to consider are how the prices change and the impact of energy sources. In our setup, we defined that the grid has nuclear and gas generation at its disposal. Then, to compute the price of energy and its climate impact, we used the following parameters: nuclear energy rate ($r_t^{\text{nuclear}}$), nuclear energy emission factor ($c_t^{\text{nuclear}}$), gas energy rate ($r_t^{\text{gas}}$) and gas energy emission facto ($c_t^{\text{gas}}$). Following the real behavior of these two generation technologies, we assume that nuclear maintains constant power during the day, defined as $E^{\text{nuclear,ratio}}$ and that the gas generation has a planned generation profile that can also be inputted as a parameter.

