# OpenReview forum: "FRESCO: Federated Reinforcement Energy System for Cooperative Optimization"
_ICLR.cc/2023/TinyPapers — Submitted to Tiny Papers @ ICLR 2023_

### Official Review · Reviewer_td3n · 2023-03-19

**Confidence:** 4

**Summary Of Contributions:**

A good application work but there are some issues in RL concepts.

**Rating:**

Great Start (GS): a submission which meets some of the reviewing criteria but has room for improvement

**Strengths And Weaknesses:**

Summary:

The authors propose a framework named FRESCO to easily control the energy grid. This framework has a three-layer architecture for hierarchical micro-gird control, uses reinforcement learning to control energy gird and federated learning scheme to train RL agents.


Strength:

The paper is well-organized and easy to follow. The experimental details are enough for this limited space. As an application work on energy grid, the experimental results have shown the proposed framework outperforms the baseline methods.

Weakness:

The authors may confuse the concepts of single-agent RL and multi-agent RL. There are three-layers architecture and each layer’s entity should be an agent. Therefore, to enable cooperation between these three layers, the authors are supposed to adopt multi-agent framework instead of single-agent framework. Besides, if the adopted algorithm is A2C, how can agent utilize extra state and communication information?

There are some over-claim for this proposed framework, the authors should reconsider these claims. For example, in the first sentence of conclusion: “FRESCO framework demonstrates how a federated reinforcement learning (FRL) approach can enable effective communication and cooperation among entities as smart grids move towards decentralization while keeping private the end user’s private information.” In fact, based on the current paper, it’s hard to see the FRL method can enable effective communication and cooperation. The only experiments have nothing to do with communication and cooperation.

**Suggested Changes:**

I would recommend the authors to future explain how the current framework enable coordinations and communications.

---

### Official Review · Reviewer_T24T · 2023-03-30

**Confidence:** 4

**Summary Of Contributions:**

This work introduces a hierarchical control system for energy grids using a combination of federated learning and reinforcement learning to train control agents. The work aims to enhance the implementation of efficient and effective energy market toward green energy.

**Rating:**

Great Start (GS): a submission which meets some of the reviewing criteria but has room for improvement

**Strengths And Weaknesses:**

Strengths:
- The work demonstrates a great vision and succinctly communicates its objectives and plan of execution.

Weaknesses:
- While training data was generated synthetically, very little information was shared on how it was generated and what the synthetic data entails. Also, there is no information on the non-i.i.d nature of the data, as should be expected for a federated reinforcement learning system simulating real-world scenarios.
- Federated learning currently has issues with communication, especially with how “merging” disparate updates from different workers through federated averaging heavily degrades model performance due to the heterogeneous nature of the data on which the workers are trained. Hence, a federated learning system beating a non-FL system seems quite skeptical. Performing and communicating hyperparameter sweep runs may put this skepticism to rest.

**Suggested Changes:**

Aside from the highlighted weaknesses,

- Figure 1 could be enlarged in scale. Currently, it is tiny and communicates very little information.
- Figures 2 and 3 look aesthetically pleasing, but it isn't apparent if they are supposed to be under an Appendix section or section 3. Moreover, there was no reference to either figure in the write-up, which makes it hard to figure out the insights they communicate.

---

### Meta-Review · Area_Chair_pYk9 · 2023-04-07

**Recommendation:** Invite to revise
**Confidence:** 4

**Metareview:**

The paper introduces a hierarchical control system for energy grids that uses a combination of federated learning and reinforcement learning to train control agents, with the aim of enhancing the implementation of efficient and effective energy markets towards green energy. The paper has strengths in terms of having a clear vision and plan of execution, but weaknesses in terms of insufficient information on training data generation and skepticism towards the use of federated learning. The suggested changes include enlarging Figure 1 and providing more information on the insights communicated by Figures 2 and 3. The authors may also need to reconsider their claims about the framework's ability to enable effective communication and cooperation among entities and adopt a multi-agent framework for cooperation between the layers.

**Summary:**

The paper proposes a green energy control system with federated and reinforcement learning, but has weaknesses and may need a multi-agent framework.

**Comments And Feedback To The Authors:**

Thank you for submitting your paper on a hierarchical control system for energy grids using federated and reinforcement learning. Your work has some strengths, including a clear vision and plan of execution, a well-organized presentation, and experimental results that show the proposed framework outperforms baseline methods.

However, there are some weaknesses in terms of insufficient information on training data generation and skepticism towards the use of federated learning. Additionally, there are issues with the concepts of single-agent RL and multi-agent RL, and some over-claims about the proposed framework's ability to enable effective communication and cooperation among entities.

To address these weaknesses, I recommend :

1. Provide more information on training data generation and the non-i.i.d nature of data to address the weakness in data generation.

2. Clarify the concepts of single-agent RL and multi-agent RL and adopt a multi-agent framework for cooperation between the layers to address the issues with the concepts.

3. Consider performing and communicating hyperparameter sweep runs to put skepticism towards federated learning to rest.

4. Reconsider the over-claims about the framework's ability to enable effective communication and cooperation among entities.

5. Enlarge Figure 1 and provide more information on the insights communicated by Figures 2 and 3.

I also suggest enlarging Figure 1 and providing more information on the insights communicated by Figures 2 and 3.



**Reason For Not Giving A Higher Recommendation:**

The paper has weaknesses in terms of insufficient information on training data generation and skepticism towards the use of federated learning, and there are issues with the concepts of single-agent RL and multi-agent RL. Additionally, there are over-claims about the proposed framework's ability to enable effective communication and cooperation among entities. These factors contribute to the decision not to give a higher recommendation.

**Reason For Not Giving A Lower Recommendation:**

N/A

---

### Decision · Program_Chairs · 2023-04-09

Revision accepted; invite to archive